# Genetic variation drives seasonal onset of hibernation in the 13-lined ground squirrel

Katharine R. Grabek [1,2]*, Thomas F. Cooke[3,4], L. Elaine Epperson [5], Kaitlyn K. Spees[3], Gleyce F. Cabral [3,6], Shirley C. Sutton [7], Dana K. Merriman [8], Sandra L. Martin[9] & Carlos D. Bustamante[1,10]*

Hibernation in sciurid rodents is a dynamic phenotype timed by a circannual clock. When housed in an animal facility, 13-lined ground squirrels exhibit variation in seasonal onset of hibernation, which is not explained by environmental or biological factors. We hypothesized that genetic factors instead drive variation in timing. After increasing genome contiguity, here, we employ a genotype-by-sequencing approach to characterize genetic variation in 153 ground squirrels. Combined with datalogger records ($n = 72$), we estimate high heritability (61–100%) for hibernation onset. Applying a genome-wide scan with 46,996 variants, we identify 2 loci significantly ($p < 7.14 \times 10^{-6}$), and 12 loci suggestively ($p < 2.13 \times 10^{-4}$), associated with onset. At the most significant locus, whole-genome resequencing reveals a putative causal variant in the promoter of FAM204A. Expression quantitative trait loci (eQTL) analyses further reveal gene associations for 8/14 loci. Our results highlight the power of applying genetic mapping to hibernation and present new insight into genetics driving its onset.

[1] Department of Genetics and Department of Biomedical Data Science, Stanford University School of Medicine, Stanford, CA, USA. [2] Fauna Bio Incorporated, Berkeley, CA, USA. [3] Department of Genetics, Stanford University School of Medicine, Stanford, CA, USA. [4] Whitehead Institute for Biomedical Research, Cambridge, MA, USA. [5] Center for Genes, Environment and Health, National Jewish Health, Denver, CO, USA. [6] Laboratório de Genética Humana e Médica, Universidade Federal do Pará, Rua Augusto Corrêa, 1 - 66.075-110, Belem, PA, Brazil. [7] Department of Genetics and Department of Cardiovascular Medicine, Stanford University School of Medicine, Stanford, CA, USA. [8] Department of Biology, University of Wisconsin Oshkosh, Oshkosh, WI, USA. [9] Department of Cell and Developmental Biology, University of Colorado School of Medicine, Aurora, CO, USA. [10] Chan Zuckerberg Biohub, San Francisco, CA, USA. *email: krgrabek@gmail.com; cdbustam@stanford.edu

Hibernation is a highly dynamic phenotype that maximizes energy savings during periods of low resource availability. For sciurid rodents, including the 13-lined ground squirrel, *Ictidomys tridecemlineatus*, an endogenous circannual clock entrained by light controls the timing of winter hibernation[1–3], along with rhythms in reproductive behavior, food intake, and body mass[4–6]. These hibernators partition their year between two distinct states, homeothermy and heterothermy (a.k.a hibernation, Fig. 1a) that are distinguished by dramatic differences in behavior and physiology. While physiology during homeothermy resembles that of a non-hibernating mammal, ground squirrels spend most of their hibernation time in an energy-conserving state called torpor (Fig. 1b). Here, metabolic, respiratory and heart rates are dramatically reduced to 1–9% of homeothermic baselines while body temperature is lowered to

near freezing[7]. However, torpor is not continuous but instead punctuated by brief, metabolically intense arousals that largely restore baseline physiology, including near-homeothermic body temperature[8]. Thus, hibernation is a period of heterothermy composed of cycles between torpor and arousal.

The seasonal transition from homeothermy to heterothermy occurs during the autumn of each year. Successful hibernation requires preparation, most notably the storage of large amounts of energy in the form of fat, because this species fasts throughout the heterothermic period. While post-reproduction homeothermy is marked by increased food intake, as the onset of heterothermy approaches, the squirrel's metabolic rate slows, peak body mass is achieved, and food intake ceases[9]. At the cellular level, glucose-based metabolism is switched to one that is primarily lipid-based, and lipogenesis is swapped for lipolysis[10]. While peak plasma

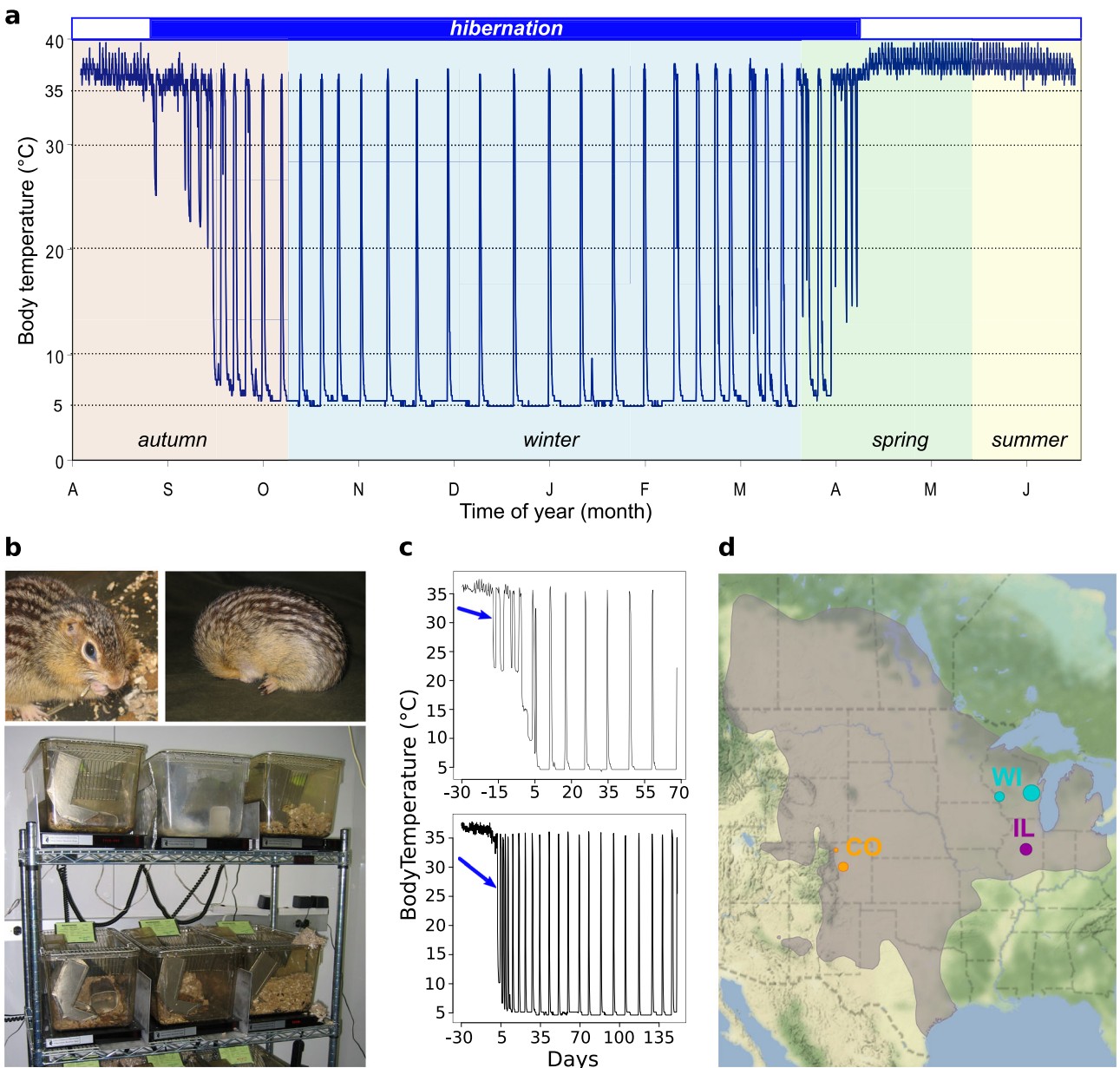

**Fig. 1 The 13-lined ground squirrel as a model for studying the genetics of hibernation. a** Body temperature trace showing a 13-lined ground squirrel's year. Hibernating portion is demarcated by blue shaded box above. **b** A non-hibernating and hibernating 13-lined ground squirrel, individually housed in standard lab rodent cages in an animal facility. **c** Representative plots of body temperature telemetry analyses. Arrows point to the first day of torpor, the phenotype measured in this study. Days are ± from hibernaculum placement. **d** Approximate locales of squirrels genotyped in this study. The taupe shaded area indicates the 13-lined ground squirrel's geographic range with data provided by NatureServe[76]. The map was created using ggmap[77] for R.

insulin concentration occurs during this period, paradoxically animals also become transiently insulin-resistant[11]. At the mRNA and protein levels, sweeping changes in expression are observed[12,13]. However, the genetic factors driving this transition remain largely unknown.

The commencement of torpor, defined here by a criterion drop in body temperature, is one readily quantifiable outcome of the transition that marks the start of seasonal heterothermy. When housed under standard laboratory conditions in an animal facility (Fig. 1b), 13-lined ground squirrels exhibit individual variation in the timing of their first bout of torpor. This variation is not accounted for by the environmental signals of food withdrawal, shortened photoperiod, or falling ambient temperature, or by the biological factors of age, body mass, and sex. All of these variables have little to modest influence on timing[14]. Rather, consistent with being controlled by an endogenous circannual clock, we hypothesized that observed variation in the onset of torpor is due to underlying genetic variation between individuals. If so, applying a genome-wide scan has the potential to identify genetic components driving the start of seasonal heterothermy.

Therefore, in this study, we first increased the contiguity of the 13-lined ground squirrel draft genome assembly. We next employed a genotype-by-sequencing strategy to characterize genetic variation in 153 13-lined ground squirrels whose tissues were previously collected for use in biochemical, transcriptomic and proteomic studies[15–17]. Many of these squirrels were surgically implanted with body temperature dataloggers, and from their records, we recorded the first day that torpor occurred in each individual (Fig. 1c). We next estimated the heritability of, and identified genetic variants associated with, the onset of autumn torpor in this species. Finally, we integrated data from prior transcriptomic studies to identify transcripts whose expression levels were significantly associated with these variants. Our results present new insight into the genetics driving the transition from homeothermy to heterothermy in a mammal and illustrate the power of genetic analysis to attack questions of exceptional biological significance in a non-classical genetic model organism.

## Results

**Long-range scaffolding of the draft genome assembly.** At the time this study began, the existing 13-lined ground squirrel genome assembly, like that of many non-model organisms, contained thousands of unordered scaffolds, which could lead to difficulties in identifying causative variants, as peaks in linkage disequilibrium (LD) could be spread across multiple scaffolds. We therefore first sought to increase the genome's contiguity using a long-range scaffolding technique[18] from Dovetail Genomics.

A single library was constructed using proximity ligation of in vitro reconstituted chromatin. After sequencing, which provided 52.6× physical coverage of the genome (Table 1) and scaffolding, the contiguity of the final HiRise assembly was increased approximately three-fold as compared to the existing draft assembly (N50 of 22.6 Mb vs. 8.19 Mb; Table 1, Supplementary Fig. 1 and Supplementary Data 1). The longest scaffold increased from 58.28 Mb to 73.92 Mb. Importantly, 539 original draft assembly scaffolds were reduced to just 33 scaffolds, which now contained half of the genome (Supplementary Fig. 2).

**Identification of genetic variants.** We next applied a modified ddRAD sequencing protocol previously described in ref. [19] to generate libraries for 153 13-lined ground squirrels from which we obtained DNA from frozen tissue. After aligning the resulting sample library reads to the HiRise genome assembly (Supplementary Data 2), we retained 337,695 loci (50.65 Mbp)

**Table 1 The HiRise assembly increases the contiguity of the SpeTri2.0 draft genome assembly.**

| Assembly details | SpeTri2.0 | HiRise |
|---|---|---|
| Average coverage | 495.1× | 52.6× |
| Total length (Mb) | 2,478.4 | 2,478.4 |
| Contigs | | |
| No. of contigs | 153,485 | 153,521 |
| Contig N50 (kb) | 44.137 | 44.131 |
| Scaffolds | | |
| No. of scaffolds (≥0) | 12,483 | 10,007 |
| Longest scaffold (Mb) | 58.28 | 73.93 |
| Scaffold N50 (Mb) | 8.19 | 22.6 |
| No. of scaffolds ≥ N50 | 80 | 33 |
| Scaffold N90 (Mb) | 1.13 | 3.33 |
| Gaps | | |
| Number of gaps | 141,005 | 143,517 |
| Percent of genome in gaps | 6.75% | 6.76% |

that fell between predicted BglII and DdeI target regions, with coverage of at least one read in one individual. Applying variant calling and filtering to these loci (Supplementary Fig. 3), we next identified 786,453 biallelic variants, which had an overall Ti/Tv ratio of 2.19, comparable to ratios reported within intronic and intergenic regions[20], and more specifically, exonic regions of the 13-lined ground squirrel genome[21]. For use in downstream analyses, we retained 575,178 variants for which genotypes were present in at least 90% of the individuals. Of these retained variants, 35,257 were indels, whereas 539,921 were single nucleotide polymorphisms.

**Population structure and genetic relatedness.** The squirrels genotyped in this study originated from wild stock trapped in disparate geographical locales (Fig. 1d). The records for their exact source and relatedness were not always available. This was not due to intentional sampling design, but rather due to the availability of squirrels each year, either trapped from the wild or supplied from a breeding colony, and the biological questions originally being pursued. Therefore, to identify population structure within our sample set, we applied ADMIXTURE clustering with 5-fold cross-validation[22] on $K = 2$ through $K = 10$ ancestral populations using a set of 54 unrelated individuals who best represented the ancestries of all squirrels (Supplementary Fig. 4; see Methods). We then applied ADMIXTURE projection to estimate proportions of learned ancestries within the remaining 99 squirrels. The lowest cross-validation error occurred at $K = 3$ (Fig. 2a), where individuals separated into Colorado (CO), Illinois (IL), and Wisconsin (WI) components. The pairwise genetic distance ($F_{ST}$) estimates between populations were 0.47 and 0.31 for CO vs. WI and IL, respectively, and 0.30 for WI vs. IL, indicating moderate to strong genetic drift.

At $K = 6$, we observed separation most consistent with records about sampling (Fig. 2a). For instance, the algorithm identified a La Crosse, WI (LaX) ancestral component for the squirrels supplied from the UW Oshkosh breeding colony in 2010, matching the breeding records for that year. Additionally, the algorithm identified two ancestral components for the IL squirrels: those purchased in 2006 ('06) belonged to a single ancestry, while those from 2010 ('10) segregated into another ancestry, suggesting different trapping locales between years. While records about the origins of the UW Oshkosh squirrels supplied prior to 2010 were unavailable, the algorithm identified two ancestral components for this breeding colony. The pairwise $F_{ST}$ values were still consistently high (0.33–0.48) among all populations (Supplementary Table 1), except for the two

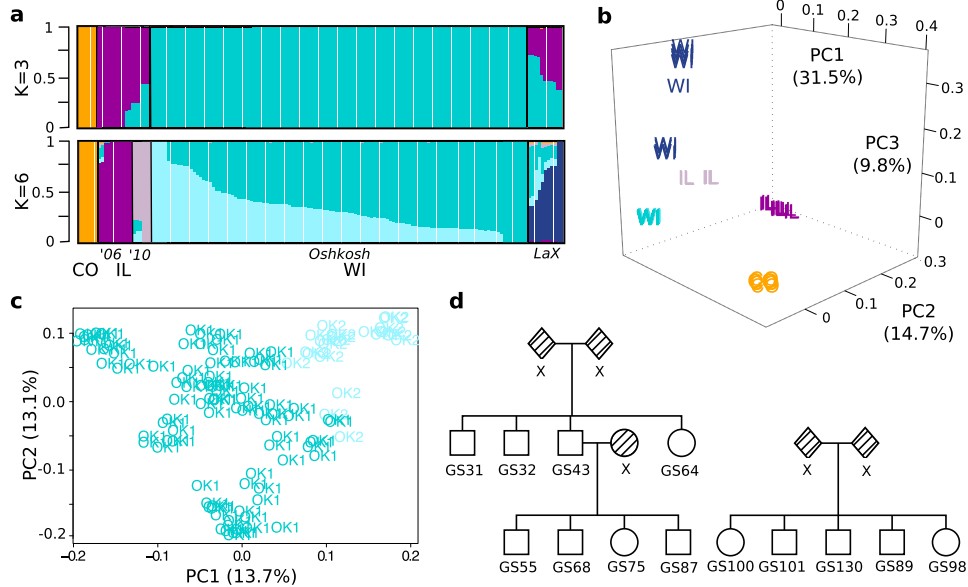

**Fig. 2 Genotype-by-sequencing reveals population structure and relatedness among sampled squirrels. a** ADMIXTURE analysis results showing $K = 3$ or $K = 6$ genome-wide specific ancestry estimates. Squirrels are shown as vertical bars with proportion of specific ancestry colored within each bar. Populations are clustered and labeled by geographic sampling locales (U.S. state, and for WI, city) and for those from IL, sampling years. **b** Principal components analysis of all 153 genotyped squirrels. The first 3 PCs are plotted, with individuals labeled by state and colored by the population for which they have the greatest proportion of ancestry as determined by $K = 6$ ADMIXTURE analysis shown in **a**. **c** Principal components analysis of 119 squirrels from the Oshkosh WI population. The first two PCs are plotted with individuals labeled and colored by Oshkosh sub-population (*OK1* or *OK2*) as determined by $K = 6$ ADMIXTURE analysis shown in **a**. **d** Representative pedigrees reconstructed from identity by descent (IBD) and kinship coefficient estimates of the Oshkosh squirrels. Shaded shapes labeled x indicate relatives not genotyped in this study.

**Table 2 Timing of the first of torpor bout in autumn is heritable.**

| Trait | Sample size (*n*) | Method | Fixed effects | Fixed effects $R^2$ | Genetic variance | Error variance | Genetic proportion | C.I. (95%) |
|---|---|---|---|---|---|---|---|---|
| Timing of torpor onset | 72 | LM | Sex, year, $T_a$ | 0.2505 | — | — | — | — |
| | | LMM | | — | 57.06 | 0.00 | 1 | 1–1 |
| | | MCMCgrm | | — | 49.8 | 0.41 | 0.99 | 0.61–0.99 |

$T_a$, ambient temperature $\leq 14\,°C$, denotes day of placement into hibernaculum

(non-LaX) within Oshkosh (0.16) and the two within IL (0.23), again supporting the notion of limited gene flow at increased geographical distance.

The first three principal components (PCs) from a PCA-based analysis recapitulated both the observed ADMIXTURE $K = 6$ clustering and the known geographical sampling locales of the squirrels (Fig. 2b). All populations were distinctly separated, except for the two within Oshkosh, whose separation was only observed at the higher PCs (PC17 and PC19, Supplementary Fig. 5).

**Genetic relatedness within the Oshkosh breeding colony**. Due to the strong population structure, we limited further analysis to just the Oshkosh group of squirrels (not including LaX, $n = 119$), for which we were able to collect the most phenotypic measurements ($n = 72$, Supplementary Data 3) from analysis of the body temperature telemetry data, as opposed to fewer than $n = 10$ phenotypic measurements in each of the remaining populations. These squirrels were largely admixed between two Oshkosh sub-populations; however, individual proportion of ancestry was not significantly correlated with the timing of hibernation onset ($r = 0.26$, $p = 0.24$, Supplementary Fig. 6).

Our existing records from the breeding colony suggested that many of these squirrels were littermates, although exact relatedness was unknown. We therefore estimated relatedness, adjusting for population substructure with the first PC[23,24], which distinguished the two ancestral components (Fig. 2c). Using pedigree reconstruction[25], we identified 19 first-degree families, to which 80% of the squirrels belonged. Consistent with our records, most of these families were composed solely of littermates (Fig. 2d), although in some cases we also identified parent-offspring relationships (Fig. 2d).

**Heritability estimates for timing of autumn torpor**. We next investigated the effect of genetic variation on autumn torpor onset within the Oshkosh subset of animals. We estimated heritability of this trait using a linear mixed model, in which we controlled for sex, year of monitoring and date of placement into the hibernaculum (our fixed effects, see methods), and we input the genetic relatedness estimates as the random effect. Unexpectedly, this model converged with no residual error in the variance components, resulting in an estimate of 100% heritability (LMM, Table 2). To confirm this high estimate, we fit a separate Bayesian multivariate general linearized mixed model

with the same fixed and random effects. Here, the posterior mode of heritability was 99%, and the confidence intervals were between 61 and 99.9% (MCMCgrm, Table 2); even the lower bound of the estimate still indicated high heritability, confirming our hypothesis that underlying genetic factors drive variation in the onset of torpor.

**Genome-wide association scan.** We identified genetic loci associated with the onset of autumn torpor by performing a genome-wide association scan (GWAS) using 46,996 variants with a minor allele frequency (MAF) ≥0.05 and the fit from the linear mixed model. After applying a Bonferroni correction to adjust for 14,004 estimated independent tests[26], our significance threshold at $\alpha = 0.1$ was $p < 7.14 \times 10^{-6}$. We observed two separate loci significant for genome-wide association, SNP 1 ($p_{\text{unadjusted}} = 3.81 \times 10^{-6}$, $p_{\text{Bonferroni}} = 0.053$) and SNP 2 ($p_{\text{unadjusted}} = 3.90 \times 10^{-6}$, $p_{\text{Bonferroni}} = 0.054$; Table 3).

In addition to these two genome-wide significant loci, the Manhattan plot showed several other distinct peaks (Fig. 3a), indicating potential associations of other loci with the timing of torpor onset. A plot of the observed vs expected quantiles of log-transformed $p$-values (Q–Q plot) also exhibited an excess of significant values well above the red line in the tail of the distribution (Fig. 3b). In contrast, results from 5000 rounds of GWAS tests with permuted phenotypes revealed no excess of $p$-values above the null distribution (Supplementary Fig. 7). To identify additional loci potentially associated with torpor onset, we set an exploratory $p$-value threshold of $p < 2.13 \times 10^{-4}$, where we expected 10 variants to be randomly associated under the null model. Instead, we observed 42 associated variants. We further estimated the probability of observing 42 variants with permutation testing: after each round of GWAS permutation, we counted the number of variants reaching the exploratory (i.e., suggestive significance) $p$-value threshold and generated an empirical null distribution of observed variant counts from 5000 permutations (Supplementary Fig. 8). Similar to our expectations, the median number of variants meeting suggestive significance was 9. In contrast, we observed counts of 42 variants only in the top 2% of permutation tests; in other words, the probability for observing this number of suggestively associated variants under the null distribution was $p = 0.02$.

After accounting for LD among the 42 variants associated with torpor onset, we identified 12 independent loci suggestively associated with the phenotype (SNP 3–SNP 14, Fig. 3a and Table 3). While the estimated mean allelic effect size for all variants was −0.04 days (SD = 1.84, $n = 46{,}996$), the effect sizes for the 2 genome-wide significant and 12 suggestive variants were all within either the top or bottom 1% of the total distribution, being at least ±4.25 days for each additional allele (Fig. 3c and Table 3).

The genome-wide inflation factor lambda ($\lambda$) for the GWAS was 0.979, indicating we had adequately corrected for population structure. While this is often adjusted via its input as a fixed effect covariate in the linear mixed model (e.g., the Q matrix), genetic relatedness alone (e.g., the random effect K matrix) may also sufficiently account for population structure[27]. We observed this to be the case in our model; a plot of the cumulative distribution function of the $p$-values revealed no upward skew (Supplementary Fig. 9), whereas adding the Q matrix resulted in a global deflation of $p$-values ($\lambda = 0.91$, Supplementary Fig. 9).

We next estimated the amount of phenotypic variance explained by the 2 significant and 12 suggestive loci. We used linear regressions to assess variance explained by combinations of multiple loci, with or without the known fixed effects. While the initial model fit with just the fixed effects of sex, year of monitoring and date of hibernaculum placement accounted for 25% of the variance in onset of torpor (Fig. 4a), the 14 loci alone explained 48.4% of the variance (Fig. 4b) and when combined with the fixed effects, accounted for 79.5% of the total variance in the phenotype (Fig. 4c). Furthermore, the subset of the top five most significant loci (SNP 1–SNP 5, Table 3) explained 39% of the variance, excluding fixed effects. Hence, a relatively small subset of loci accounted for most of the genetic component underlying the timing of autumn torpor in this population of 13-lined ground squirrels.

When we examined the genes located nearest to the significant and suggestive loci, we observed themes functionally consistent with physiology underlying the transition to hibernation, such as insulin processing and signaling, feeding and satiety, and control of heart rate (Table 3). Intriguingly, the top genome-wide significant variant, SNP 1, was located nearest the gene *family with sequence similarity 204 member A* (*FAM204A*), whose function is poorly characterized (Fig. 5a). The second genome-wide significant marker, SNP 2, was located between two genes that are also functionally relevant within the scope of hibernation: *coiled-coil-helix-coiled-coil-helix domain containing 3* (*CHCHD3*) and *exocyst complex component 4* (*EXOC4*; Fig. 5b). While *CHCHD3* maintains the structural integrity of mitochondrial cristae[28], *EXOC4* is a component of the exocyst complex involved in the secretion of insulin[29], as well as lipid and glucose uptake in response to insulin signaling[30]. SNP 3, the most significant variant within the suggestive subset of loci, was closest to *motilin* (*MLN*; Supplementary Fig. 10a), a small peptide hormone that regulates gastrointestinal contractions and stimulates hunger signaling[31]. SNP 5 was located within the intron of the *muscarinic acetylcholine receptor M2* (*CHRM2*, Supplementary Fig. 10b). This receptor mediates bradycardia in response to parasympathetic-induced acetylcholine release[32], a phenomenon that occurs during entrance into torpor[33]. SNP 10 (Supplementary Fig. 10c), weakly-linked ($r^2 = 0.33$) and located ~500 kb from SNP 5, was located near *pleiotrophin* (*PTN*), a growth factor involved in neurogenesis, axonal outgrowth, angiogenesis and hematopoiesis of bone marrow[34]. Finally *prohormone convertase 2* (*PCSK2*), an enzyme that activates hormones and neuropeptides including cleavage of proinsulin into its mature form[35], was located in close proximity to SNP 14 (Supplementary Fig. 10d).

**Corroboration of significant and suggestive loci.** Because no additional dataset was available to independently replicate the findings from our GWAS, we used an alternative published method, denoted ComPaSS-GWAS, where samples are split into complementary halves multiple times and GWAS are performed on the halves in each resample[36]. To compensate for small sample size, we randomly separated samples into two subsets of size $n = 48$, allowing 24 samples to overlap and distributing the remaining 48 samples into two complementary ($n = 24$) halves. With a critical value of $\alpha = 1 \times 10^{-3}$, corroboration parameter $\eta = 0.6$ and 100 resamples, the ComPaSS-GWAS corroborated both SNP 1 ($\eta = 0.66$) and SNP 2 ($\eta = 0.77$) as being significantly associated with the onset of autumn torpor, as well as four of the suggestively associated variants: SNP 3 ($\eta = 0.76$), SNP 7 ($\eta = 0.66$), SNP 8 ($\eta = 0.66$) and SNP 10 ($\eta = 0.70$). For the remaining suggestive loci, the corroboration values ranged from $\eta = 0.18$–$0.54$ (Table 3). In contrast, when we examined 1000 additional variants randomly sampled across the distribution of effect sizes, none were corroborated for association with the phenotype. The median value for this subset was $\eta = 0.01$, and only three variants exceeded $\eta = 0.18$, the lowest corroboration value observed among all of the significantly and suggestively associated loci (Supplementary Fig. 11).

**Table 3 GWAS identifies variants significantly (SNPs 1 & 2) and suggestively (SNPs 3–14) associated with the onset of torpor.**

| SNP # | HiRise | | SpeTri2.0 | | Ref | Alt | MAF | p-value | β | Candidate gene(s) | η |
|---|---|---|---|---|---|---|---|---|---|---|---|
| | Scaffold | Position | Scaffold | Position | | | | | | | |
| 1 | Scyvm7L_301 | 9767851 | JH393296.1 | 7909312 | C | T* | 0.14 | 3.81E-06 | 7.32 | FAM204A | 0.66 |
| 2 | Scyvm7L_2912 | 2711274 | JH393389.1 | 2499372 | A | C* | 0.46 | 3.90E-06 | 5.10 | EXOC4; CHCHD3 | 0.77 |
| 3 | Scyvm7L_146 | 3843670 | JH393286.1 | 25104888 | G* | A | 0.38 | 7.79E-06 | 5.37 | MLN | 0.76 |
| 4 | Scyvm7L_301 | 17167980 | JH393296.1 | 15309441 | C | T* | 0.19 | 2.77E-05 | 5.75 | FOXI2; NPS | 0.54 |
| 5 | Scyvm7L_4270 | 16503923 | JH393402.1 | 1439665 | G | T* | 0.37 | 3.20E-05 | −5.08 | CHRM2 | 0.45 |
| 6 | Scyvm7L_146 | 3229252 | JH393286.1 | 25719306 | A | G* | 0.16 | 3.70E-05 | −7.45 | HLA-DPB1 | 0.48 |
| 7 | Scyvm7L_2116 | 23692482 | JH393326.1 | 1868477 | C* | T | 0.42 | 4.73E-05 | 4.27 | MFAP3L; CLCN3 | 0.66 |
| 8 | Scyvm7L_9858 | 615414 | JH393647.1 | 629692 | G* | A | 0.37 | 4.92E-05 | 5.42 | MUC21; DDR1 | 0.66 |
| 9 | Scyvm7L_100 | 29226048 | JH393613.1 | 919871 | G* | A | 0.31 | 6.02E-05 | −4.71 | CDK6; SAMD9 | 0.39 |
| 10 | Scyvm7L_4270 | 16068966 | JH393402.1 | 1874622 | C* | A* | 0.48 | 6.19E-05 | 5.18 | DGKI; PTN | 0.70 |
| 11 | Scyvm7L_1164 | 849226 | JH393548.1 | 1324176 | G | A | 0.10 | 9.33E-05 | −7.95 | HPSE; SCD5 | 0.41 |
| 12 | Scyvm7L_936 | 9851454 | JH393624.1 | 94391 | G | A*+ | 0.44 | 1.33E-04 | −5.33 | DKK2 | 0.28 |
| 13 | Scyvm7L_100 | 29569341 | JH393613.1 | 1263164 | G | T* | 0.34 | 1.54E-04 | 4.86 | FAM133B | 0.45 |
| 14 | Scyvm7L_1707 | 29969722 | JH393295.1 | 2605622 | G* | C | 0.17 | 1.80E-04 | −5.44 | PCSK2 | 0.18 |

Ref is the allele reported in the reference genome assembly. * Allele for which effect size is estimated; minor allele unless otherwise specified with + . + Major allele. MAF is the minor allele frequency estimated from the n = 72 genotypes used in the association scan. β is the effect size estimate in days. Candidate gene(s) are genes nearest to variant; additional genes listed may be functionally related to hibernation. η is the corroboration estimate from ComPaSS-GWAS[36]

**Identification of a putative causal variant at SNP 1 locus.** We next sought to identify the putative causal variant within the most significantly associated locus, SNP 1. We resequenced whole genomes of 12 individual squirrels, 6 each from the early and late torpor onset groups, at approximately 9x coverage and identified 944 variants linked to SNP 1. Of these, 900 variants were intergenic, 42 were intronic in *FAM204A* and 1 was located in the 3'UTR of *FAM204A*. The remaining variant (Scyvm7L_301:9,837,692 or SpeTri2.0 JH393296:7,979,153, G > A) was located approximately 13 bp upstream of the transcriptional start site of *FAM204A* (Fig. 5c) and within a region conserved across mammals, including distantly-related hibernators (Fig. 5c, d). The base itself had a GERP++ score of 4.63, indicating it is under evolutionary constraint[37], yet significantly, the derived allele (A) was the major allele in this population of ground squirrels and therefore associated with earlier torpor onset. An in silico analysis predicted this variant to flank the CCAAT box within an NF-Y(A/B) transcription factor binding site[38] (Fig. 5c). Although epigenetic and transcription factor binding-site data is largely unavailable for the 13-lined ground squirrel, in humans, ENCODE data shows the homologous base (hg19 chr10:120,101,838) within the promoter of *FAM204A*, and more specifically, within an NF-Y(A/B) transcription factor binding site (Supplementary Fig. 12). Additionally, the FATHMM-MKL score[39] for this variant was 0.989, predicting it to be deleterious in humans. In sum, this variant putatively disrupts a conserved transcription factor binding site in the promotor of *FAM204A* to potentially alter its expression.

**Identification of eQTLs using transcriptomic datasets.** After identifying a variant in the promoter of *FAM204A*, we hypothesized that other significant and suggestive loci would also be linked to gene regulatory variants. We therefore applied an expression quantitative trait loci (eQTL) analysis using the EDGE-tag transcript datasets from heart, liver, skeletal muscle (SkM) and brown adipose tissue (BAT)[16,40], as a subset of the squirrels genotyped in this study were also assayed for transcriptome expression in these prior studies. Under an additive linear model, we identified significant *cis*-eQTL associations (±500 kb, q < 0.1) for 8/14 variants (Table 4). The most significant GWAS locus, SNP 1, was also the most significant *cis*-eQTL (q < $4.4 \times 10^{-8}$), where the minor allele, linked to the non-disruptive reference allele in the *FAM204A* promoter and associated with a later onset of torpor (Fig. 6a), was correlated with increased expression of *FAM204A* in BAT (Fig. 6b; Supplementary Fig. 13a) and SkM (Table 4). Several variants were associated with expression changes in the previously identified candidate genes (Table 3). SNP 5, associated with an earlier torpor onset (Fig. 6d), was correlated with increased *CHRM2* expression in heart (Fig. 6e; Supplementary Fig. 13c), while SNP 10, associated with a later torpor onset (Fig. 6g), was also correlated with decreased expression of *PTN* in BAT (Fig. 6h; Supplementary Fig. 13e). SNP 13 correlated with increased expression of its nearest gene, *family with sequence similarity 133 member b* (*FAM133B*) in both heart and SkM (Table 4). Finally, SNP 7 associated with its two most proximal genes: here the minor allele was correlated with decreased *chloride voltage-gated channel 3* (*CLCN3*) expression in SkM, yet also correlated with increased *microfibril associated protein 3 like* (*MFAP3L*) expression in heart (Table 4).

None of the variants met significance thresholds (q < 0.1) to be identified as *trans*-eQTLs (Supplementary Data 4–7); however, this was likely due to the relatively small sample sizes (n = 22–23 in heart, SkM and liver; n = 43 in BAT) and the large number of EDGE-tags tested (25–30 K per tissue) in each dataset. We therefore examined the top *trans*-eGene (>500 kb from variant)

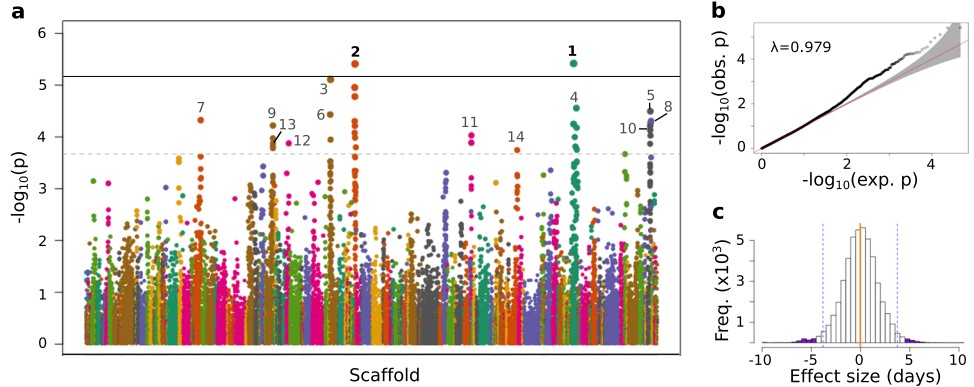

**Fig. 3 GWAS identifies genetic variants significantly associated with date of first torpor in 13-lined ground squirrels. a** Manhattan plot shows the negative log-transformed *p*-values of 46,996 variants tested for association with date of first torpor in 72 squirrels. Variants are ordered by position on each scaffold, which are colored along the *x*-axis. Solid line indicates threshold for genome-wide significance, while dashed line is threshold for suggestive significance. Numbered are significant (bold) and suggestively associated variants. Numbering is same as in Table 3 and Figs. 5 and 6. **b** Q–Q plot of the GWAS log-transformed *p*-values plotted against their expected values. **c** Histogram of effect sizes of the 46,996 variants on date of first torpor. Orange vertical line marks the mean, dashed vertical lines mark upper and lower bounds of 98$^{th}$ percentile and purple shading indicates effect sizes of the 14 significantly and suggestively associated variants.

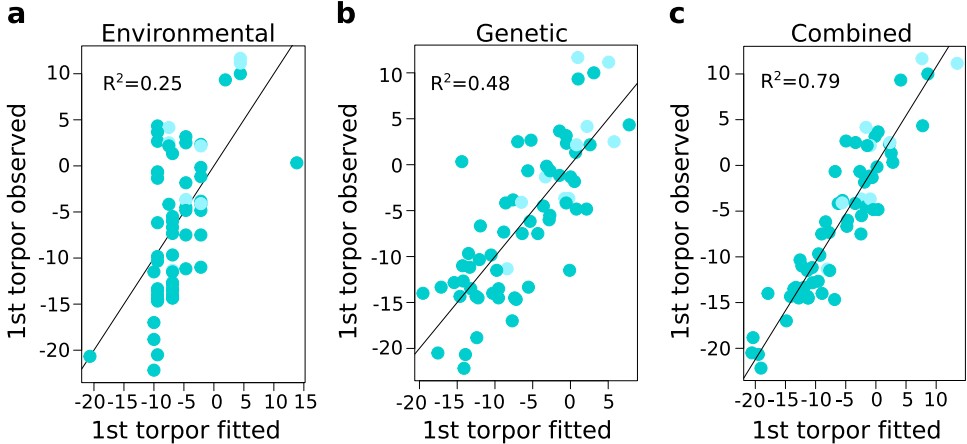

**Fig. 4 Combined environmental and genetic effects account for variation in the start of torpor.** Plots show correlation between fitted and observed values for the start of torpor (in days from hibernaculum placement) using a linear regression model with $n = 72$ independent squirrels. Adjusted R$^2$-value is labeled in each. Shading matches Fig. 2a, c. **a** Linear model fit with the environmental variables of year of monitoring and date of hibernaculum placement, and the biological variable of sex (i.e. the fixed effects, see methods). **b** Linear model fit with the 14 significant and suggestive SNP genotype combinations for each squirrel. **c** Linear model fit with the variables from both **a** and **b**.

for each significant *cis*-eQTL, hypothesizing that we would identify genes within the same pathway or consistent with the physiology of the *cis*-eGene. Indeed, in heart, the *angiotensin II receptor type 1*, *AGTR1*, involved in regulation of blood pressure[41], was the top *trans*-eGene for SNP 5 ($p = 2.97 \times 10^{-5}$, Supplementary Data 4). In contrast to *CHRM2*, this transcript showed decreased expression in relation to the minor allele (Fig. 6f; Supplementary Fig. 13d), consistent with the physiology of torpor, where reduced heart rate is coupled with decreased blood pressure[42,43]. In BAT, the top *trans*-eGene for SNP 10 was *FER Tyrosine Kinase* (*FER*; $p = 4.73 \times 10^{-5}$, Supplementary Data 7), whose expression, like *PTN*, was also decreased in relation to the minor allele of its SNP (Fig. 6i; Supplementary Fig. 13f). Both *PTN* and *FER* stimulate phosphorylation of β-catenin[34,44], suggesting a role for this pathway in the start of torpor. Finally, in BAT the top *trans*-eGene for SNP 1 was *PH domain and leucine rich repeat protein phosphatase 1* (*PHLPP1*; $p = 2.1 \times 10^{-5}$, Supplementary Data 7), whose expression, like *FAM204A*, was increased in relation to the variant (Fig. 6c; Supplementary Fig. 13b). *PHLPP1* is a protein phosphatase that

dephosphorylates and inactivates both *Akt2* and *protein kinase C*; *Akt2* is expressed highly in insulin-responsive tissues, including BAT, where it modulates glucose uptake and homeostasis[45]. Hence, *PHLPP1* gene may play a role in the switch from glucose to fat-based metabolism that occurs at the onset of seasonal heterothermy.

## Discussion

Mammalian hibernation is a highly dynamic and extraordinary phenotype that remains poorly understood. While it has been characterized at behavioral, whole body, cellular, and molecular levels, a genetic basis of the phenotype has yet to be established. Our study is the first, to our knowledge, to characterize genome-wide variation within a hibernator, the 13-lined ground squirrel. This enabled us to estimate the heritability of, and identify genetic variants associated with, the onset of seasonal heterothermy.

Our results of heritability are consistent with those from a study that reported significant heritability in spring emergence from hibernation in wild Columbian ground squirrels[46]. However, our estimates for immergence into hibernation are much

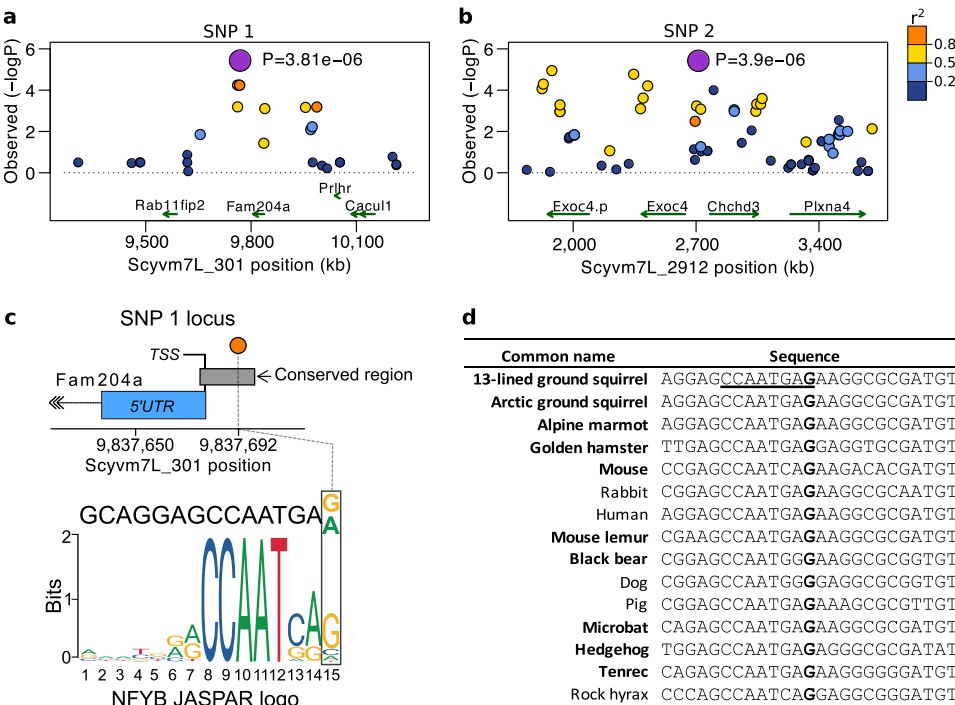

**Fig. 5 Locations of GWAS-significant loci and the putative causal variant at the SNP 1 locus. a, b** Regional Manhattan plots are centered on the GWAS-significant variants, SNP 1 (**a**) and SNP 2 (**b**), which are labeled on top by number (see Fig. 3a and Table 3) and shaded purple. Other variants within region are colored by LD value ($r^2$) in relation to the significant SNP. Genes are shown below as green arrows and labeled by gene symbol. **c** The putative causal variant linked to SNP 1 in **a** is shaded orange and located within a vertebrate conserved element, shaded gray, upstream of the transcriptional start site (TSS) of *FAM204A*, shaded blue. In silico analysis predicts the variant to disrupt position 15 of the *NFYB* motif, shown below. **d** Alignment of the 13-lined ground squirrel reference sequence to homologous sequences of other mammals, including hibernators or mammals capable of heterothermy, indicated in bold. The position of the SNP 1 putative causal variant (G > A) in **c** is in bold while sequence from positions 8–15 of the *NFYB* motif is underlined.

**Table 4 GWAS significant and suggestive variants are *cis*-eQTLs.**

| SNP # | Tissue | Tag ID | Scaffold | Position | Gene symbol | Distance (kb) | *p*-value | *q*-value | β |
|-------|--------|--------|----------|----------|-------------|---------------|-----------|-----------|------|
| 1 | BAT | Gene_99455 | Scyvm7L_301 | 9806636 | FAM204A | 38.8 | 1.8E-10 | 4.4E-08 | 1.56 |
|   | SkM | Tag_16792 |  | 9807010 |  | 39.2 | 1.6E-03 | 0.071 | 0.86 |
| 3 | BAT | Gene_92836 | Scyvm7L_146 | 3382605 | TAPBP | 461.1 | 2.2E-04 | 0.011 | −0.73 |
| 5 | Heart | Tag_114532 | Scyvm7L_4270 | 16416945 | CHRM2 | 87.0 | 1.0E-03 | 0.056 | 0.89 |
| 7 | Heart | Tag_68012 | Scyvm7L_2116 | 23593820 | MFAP3L | 98.7 | 5.5E-04 | 0.040 | 0.76 |
|   | SkM | Tag_32453 |  | 23802734 | CLCN3 | 110.3 | 1.5E-04 | 0.014 | −0.85 |
| 8 | BAT | Gene_79711 | Scyvm7L_9858 | 299681 | MRPS18B | 315.7 | 2.6E-03 | 0.080 | −0.66 |
| 10 | BAT | Gene_137408 | Scyvm7L_4270 | 16236482 | PTN | 167.5 | 1.1E-04 | 6.9E-03 | −0.67 |
| 13 | Heart | Tag_164503 | Scyvm7L_100 | 29572046 | FAM133B | 2.7 | 1.5E-07 | 3.1E-05 | 1.18 |
|   | SkM | Tag_78045 |  |  |  |  | 1.2E-03 | 0.071 | 0.91 |
| 14 | BAT | Gene_16216 | Scyvm7L_1707 | 30247643 | LOC106144723 | 277.9 | 3.6E-05 | 3.0E-03 | −0.88 |

Tag ID is from the original Edge-tag datasets[16,40]. β is effect size estimated from quantile-normalized values

higher. This is likely due to differences between monitoring animals in an animal facility, where environmental conditions and access to food are relatively constant and social cues are minimal, and monitoring animals in the field, where phenotypic plasticity in response to changing environmental conditions also influences phenological timing[47]. In the wild, differences in age (adult vs juvenile) and hibernation timing have been observed[47]. While age did not significantly affect hibernation onset in our study, it is worth noting that 74% of the squirrels in this dataset were juveniles, and therefore hibernation-naïve prior to their first torpor bout. Juveniles likely face a far greater challenge in growing and fattening sufficiently to support winter hibernation in the wild; in contrast, in a relatively constant, resource-rich laboratory environment, hibernation onset for these animals may

be particularly driven by endogenous mechanisms, thus increasing heritability estimates.

In contrast to complex human diseases, where often thousands or even millions of variants of relatively small effect influence complex physiological phenotype[48], we find that relatively few loci of large effect account for phenotypic differences in the seasonal onset of hibernation. Our results are comparable to what has been observed for adaptive traits in *A. thaliana*[49], morphological variation in domesticated dogs[50] and plate armor of threespine sticklebacks[51]. Similar to these studies, our results could be explained by local genetic adaptation[52]. Alternatively, our results may stem from limited sample size inflating the percent of explanatory variance for a single locus, a phenomenon known as the Beavis effect. Relaxing the *p*-value threshold and

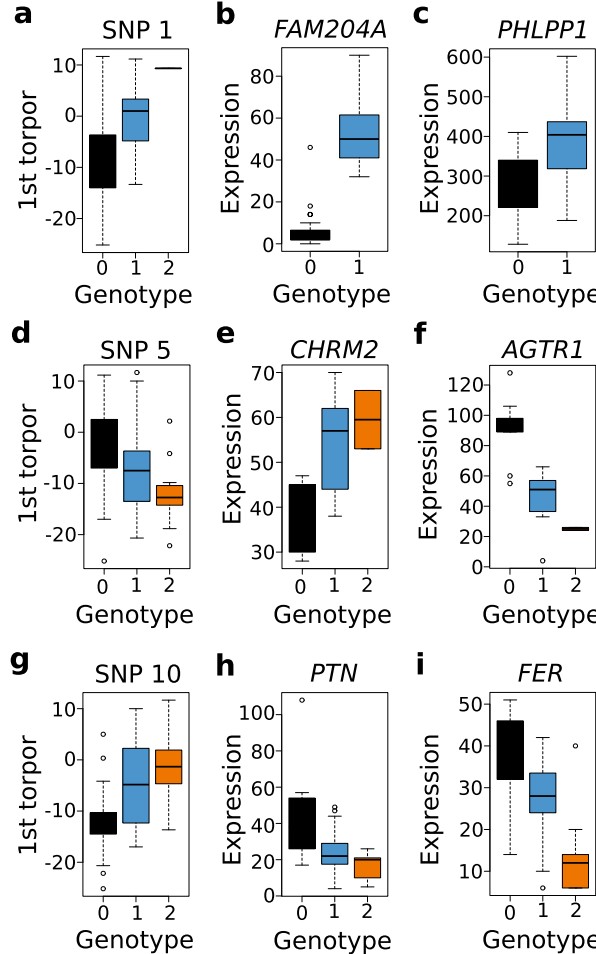

**Fig. 6 Significant and suggestive GWAS SNPs are also *cis-* and *trans-*eQTLs that explain variation in mRNA expression.** The colored boxes represent the region between the 25th and 75th percentiles, internal horizontal lines the median, outside horizontal lines the 100th percentile and open circles the outliers. Genotypes are labeled on the x-axis: 0 = major allele homozygous, 1 = heterozygous and 2 = minor allele homozygous. The mRNA expression counts, labeled on the y-axis, are taken from their original studies[16,40]. Effect of SNP 1 genotype on date of first torpor (**a**; n = 72 independent squirrels) and expression of its *cis-*eGene *FAM204A* (**b**) and *trans-*eGene *PHLPP1* (**c**) in BAT (n = 43 independent squirrels). **d–f** Effect of SNP 5 on date of first torpor, *cis-*eGene *CHRM2* and *trans-*eGene *AGTR1* in heart (n = 22 independent squirrels). **g–i** Effect of SNP 10 on date of first torpor, *cis-*eGene *PTN* and *trans-*eGene *FER* in BAT (n = 43 independent squirrels).

Within the most significantly associated locus, SNP 1, we identified a putative causal variant in the promoter of *FAM204A*. Coupled with transcriptome data[16,40], we detected that this locus was a cis-eQTL associated with *FAM204A* transcript abundance, which suggests that *FAM204A* is involved in the onset of hibernation. Intriguingly, there are few studies that have characterized *FAM204A*, especially in the context of hibernation. One recent study reported an increased number of regions surrounding *FAM204A* undergoing accelerated evolution in two species of hibernating bats and the 13-lined ground squirrel as compared to four non-hibernating mammals[54]. In addition, expression of *FAM204A* is significantly up-regulated in the brains of torpid greater horseshoe bats as compared to those that are not hibernating[55]. In humans and other non-hibernating organisms, not much is known about this gene's function. While abundance is highest in the pituitary, it is also expressed in every tissue and localizes to the nucleus (www.proteinatlas.org[56]), where it interacts with a histone acetyltransferase and a methyltransferase[57]. We therefore hypothesize that *FAM204A* may play a role in epigenetic regulation of gene expression, possibly in response to the onset of fasting and a switch to fatty acid metabolism[58], as animals prepare for hibernation in the fall.

Although more research is now needed to determine the precise role in torpor immergence for each of the remaining loci, we propose candidate genes, oftentimes the closest to the marker, due to their function being closely related to the physiology of this seasonal transition. In particular, several genes are known to modulate food intake, such as the *motilin* (*MLN*), as well as insulin signaling and processing, including *EXOC4* and *PCSK2*. Researchers have hypothesized that mechanisms governing food intake and metabolic suppression are linked, and that hibernation cannot begin until food intake has ceased[9]. Our results support this hypothesis, and present new candidates for study in hibernation.

Finally, our results highlight the power of integrating genome data with transcriptome and other high-throughput data to better understand the genetic mechanisms underlying hibernation. Prior omics screens (e.g.,[12,16,40]) have identified hundreds to thousands of genes differentially expressed among the seasonal and physiological states of the hibernator's year. While leading to insight into the pathways involved, the results of these screens do not distinguish between genes driving vs. those responding to changes in phenotype. They are also limited to the tissue and time-points being examined and may therefore miss important regulators of phenotype. For example, several *cis-* and *trans-*eGenes identified here have clear roles in the physiology of torpor, including *CHRM2* and *AGTR1*; yet neither of these were identified in the original differential expression study[40] because the timing of the first bout of torpor was not considered. Thus, by applying complementary genetic mapping approaches, current limitations inherent to gene expression screening strategies will be addressed and enable new insight into the mechanisms driving hibernation. The approaches used here can be extended to a wide variety of hibernators and the quantifiable components that comprise this highly dynamic phenotype.

increasing the number of explanatory loci will decrease the severity of this effect[53]. In light of this, we did not estimate the percent of variance explained by any single locus, instead examining estimates from combinations of loci. Although lowering the p-value threshold will also increase the false positive rate, we note that we estimated a probability of just 2% for observing the same number variants randomly associated with the phenotype at our exploratory p-value threshold. Additionally, we found that both the significantly associated variants were corroborated for association using an alternative approach for independent replication. We therefore posit that despite small sample size, both the significant loci are indeed associated with the onset of torpor. For the remaining suggestively associated loci, the results of our study now warrant independent replication with a larger sample size.

## Methods

**Animals.** Animals were procured and housed at the University of Colorado, Anschutz Medical Campus, as previously described[14]. All animal use was approved by the University of Colorado, Anschutz Medical Campus Animal Care and Use Committee.

Briefly, 130 colony-bred animals (68 females and 62 males) were obtained from the University of Wisconsin, Oshkosh in the summers of 2007–2010 (Fig. 1d, WI). These included 73 juveniles naïve to hibernation in the year of study, and 57 adults with at least one year of hibernation. While most from the colony were bred from squirrels originally wild-trapped in northeastern Wisconsin (in and around Oshkosh), several of those obtained from the Oshkosh colony in 2010 were actually

bred from either a single or both parents wild-trapped in far western Wisconsin, more than 100 miles away (in and around La Crosse, WI). However, records to identify these specific squirrels were not always maintained. In addition, 17 squirrels (9 females and 8 males; ages unknown), wild-trapped in different locales around central Illinois, were obtained from a commercial supplier (TLS Research, Bloomington, IL) in the summers of 2006 and 2010 (Fig. 1d, IL). Finally, 6 squirrels (3 females, 3 males; ages unknown) were wild-trapped in the summers of 2006 and 2009 in Elbert County and Larimer County, Colorado (5 and 1, respectively; Fig. 1d, CO).

Upon arrival, animals were housed individually in rodent cages (Fig. 1b) under standard laboratory conditions (20 ± 2 °C and 14:10 light-dark cycle, fed cat chow supplemented with sunflower seeds ad libitum). In late August or early September, animals not yet euthanized for tissue collection were surgically implanted with an intraperitoneal datalogger (iButton, Embedded Data Systems) and/or a radiotelemeter (VM-FH disks; Mini Mitter, Sunriver, OR) for remote body temperature ($T_b$) monitoring until tissue collection. The dataloggers recorded $T_b \pm$ 0.5 °C every 20, 30 or 60 min, while the radiotelemeters transmitted $T_b \pm 0.5$ °C every 20 s.

In late September or early October, the squirrels were moved to the hibernaculum to facilitate hibernation. The temperature was lowered stepwise over a two-week period to 4 °C. Food was removed as animals became torpid.

**Tissue collection and telemeter retrieval**. Liver samples were collected at different points throughout the year for use in other biochemical studies as previously described[17]. All animals were exsanguinated under isoflurane anesthesia, perfused with ice-cold saline, decapitated, and dissected on ice; tissues were immediately snap frozen in liquid nitrogen and stored at −80 °C until processed further. Telemeters were retrieved during tissue collection.

**Body temperature telemetry analysis**. To identify the first day of torpor, the telemetry data were analyzed in R[59]. $T_b$ was averaged over 4-hour windows. Homeothermic $T_b$ typically ranged from 34–39 °C. The first torpor bout was defined as the first point at which $T_b$ fell to or below 25 °C (approximately 3–5 °C above ambient prior to hibernaculum placement, Fig. 1c). Most telemetry data continuously logged $T_b$ from the beginning of September of each year. However, in several cases, telemetry recordings did not start until mid-September. Of these, only cases in which first torpor occurred after a minimum of 10 days of continuous monitoring were included for further analysis. In order to merge data across years, date of first torpor was transformed into date from placement into the hibernaculum, which was centered as day 0; hence, all days prior are negative in value, while post-placement dates are positive.

**HiRise genome assembly and annotation**. A Chicago library was prepared by Dovetail Genomics as described previously[18] from a single 100 mg frozen liver sample. Briefly, ~500 ng of high molecular weight gDNA (mean fragment length >50kbp) was reconstituted into chromatin in vitro and fixed with formaldehyde. Fixed chromatin was digested with *DpnII*, the 5′ overhangs filled in with biotinylated nucleotides, and then free blunt ends were ligated. After ligation, crosslinks were reversed and the DNA purified from protein. Purified DNA was treated to remove biotin that was not internal to ligated fragments. The DNA was then sheared to ~350 bp mean fragment size and sequencing libraries were generated using NEBNext Ultra enzymes and Illumina-compatible adapters. Biotin-containing fragments were isolated using streptavidin beads before PCR enrichment of the library. The library was sequenced on an Illumina HiSeq 2500 (rapid run mode) to produce 150 million 2 × 101 bp paired end reads, which provided 52.6× physical coverage of the genome (1–50 kb pairs).

The 13-lined ground squirrel draft assembly (SpeTri2.0), shotgun reads, and Chicago library reads were used as input data for Dovetail HiRise Scaffolding software, a pipeline designed specifically for using proximity ligation data to scaffold genome assemblies[18]. Shotgun and Chicago library sequences were aligned to the draft input assembly using a modified SNAP read mapper (http://snap.cs.berkeley.edu). The separations of Chicago read pairs mapped within draft scaffolds were analyzed by HiRise to produce a likelihood model for genomic distance between read pairs, and the model was used to identify and break putative misjoins, to score prospective joins, and make joins above a threshold. After scaffolding, shotgun sequences were used to close gaps between contigs. Supplementary Data 1 describes the input draft assembly scaffold placement within the HiRise scaffolds.

Gene annotations from the Ensembl (Release 86) and NCBI (Release 101) datasets were lifted over to the HiRise assembly using a custom Python script and Supplementary Data 1.

**Genotype-by-sequencing**. We used the modified ddRAD sequencing protocol previously described[19]. Briefly, high molecular weight DNA was extracted from 8–15 mg of frozen liver with Gentra Puregene Tissue Kits (Qiagen, Hilden, Germany). Digestion and ligation reactions were performed using 200 ng of genomic DNA from each sample with *BglII* and *DdeI* and 11-fold excess of sequencing adaptors. Samples were amplified by PCR for 8–12 cycles with a combination of index-containing primers. Between 50–60 samples were pooled in equal amounts according to their concentration of PCR product between 280–480 bp as measured

by Tapestation (Agilent Technologies, Santa Clara, CA, USA). Inserts were size-selected on a BluePippin (Sage Science, Beverly, MA) with a target range of 380 ± 100 bp and sequenced on the Illumina NextSeq in single-end 151 bp mode using a high output kit.

**Variant calling and filtering**. Reads were mapped to the 13-lined ground squirrel HiRise assembly with BWA v. 0.7.12[60]. Tables of predicted *BglII* and *DdeI* restriction digest fragments were generated as previously described[19], and sequencing coverage was measured at these sites. We then defined target regions for variant calling using the set of fragments between 125–350 bp long that had non-zero coverage in at least one individual. The mapping data are summarized in Supplementary Data 2.

Because publicly available data on 13-lined ground squirrel genetic variation is non-existent, we instead used several variant callers to identify genetic variants and to assess concordance of the genotype calls at each site. Variant calling was performed independently with Sentieon[61], Platypus[62] and Samtools[63,64]. In Sentieon, the pipeline algorithms indel realignment, base quality score recalibration, haplotyper and GVCFtyper were implemented with default settings. In Samtools, variants were called jointly using mpileup to first compute genotype likelihoods and then BCFtools to call genotypes with default parameters. Finally, variants were called jointly in Platypus with the following parameters: minFlank = 3, badReadsWindow = 5, maxVariants = 12, and minReads = 6. Only biallelic variants that both passed the filter flags and were identified by all three callers were retained (Supplementary Fig. 3). These variants were next intersected with GATK[65] and compared for genotype call concordance across samples[66]. Those that were ≥95% concordant were kept. Basic statistics about the callset, including depth, missingness, heterozygosity, Hardy–Weinberg equilibrium, and TiTv ratio were calculated in VCFtools[67]. Variants with excessive coverage (≥65×, approx. 4× the mean coverage, Supplementary Data 2) and heterozygosity (obs./exp. ratio ≥1.2) were removed from the callset (Supplementary Fig. 3). Sample libraries with excessive missingness and/or heterozygosity were removed, remade, and resequenced. Variants then were reiteratively called and filtered as described above. Finally, variants present in ≥90% of the sample libraries were used for further downstream analyses.

**Population structure and genetic relatedness estimates**. We first inferred relatedness from identity-by-state (IBS) estimates among all genotyped squirrels ($n = 153$) using KING software[68]. Due to the expectation that the Colorado squirrels are of a separate subspecies[69], relatedness was calculated independently for this subset. We selected an unrelated subset of 54 squirrels that best represented the ancestries of all squirrels within the dataset using the GENESIS package[70] in R[59]. Variants were pruned for LD in PLINK v. 1.9[71] using the parameters–indep-pairwise 50 10 0.5, which reduced the dataset to 148,870 variants. We then ran unsupervised ADMIXTURE[22] for $K = 3$ through $K = 10$ with 5-fold cross-validation. To estimate the ancestries of the remaining 99 squirrels, we ran ADMIX-TURE's projection analysis using the population structure learned in the initial unsupervised analysis, here with $K = 2$ through $K = 8$ and 5-fold cross-validation.

We performed principal components analyses (PCA) with PLINK using 90,376 LD pruned variants with MAF > 0.01 for the entire dataset and 30,356 LD pruned variants with MAF > 0.01 for the squirrels within the Oshkosh, WI, population ($n = 119$), as identified by ADMIXTURE analysis. We extracted the top 20 principal components in each analysis.

Finally, we calculated genetic relatedness among the 119 Oshkosh WI squirrels using the GENESIS package, adjusting for both population substructure and inbreeding with the first principal component[23]. We used the resulting kinship coefficients and identity-by-descent (IBD) estimates to reconstruct and visualize pedigrees among the 1st degree relatives with PRIMUS[25]. We also constructed a genetic relatedness matrix from the pairwise kinship coefficients.

**Genome-wide association scan and heritability estimates**. All analyses, unless otherwise stated, were performed in R[59]. To identify environmental and biological factors that affected the date of first torpor, we applied a linear regression using variables available from records about the squirrels. We modeled the date of first torpor as a function of the following factors: sex, year of monitoring, date of datalogger implantation, age (juvenile vs. adult), date of placement into hibernaculum and weight (as last recorded before placement into hibernaculum). We then pruned factors using stepwise regression until we identified a minimum set that did not significantly ($p < 0.05$) reduce the adjusted $R$-squared value from the initial model while minimizing AIC value. In this final model, we retained sex, year of monitoring and date of placement into hibernaculum as fixed effects.

We carried out a genome-wide association scan (GWAS) on the date of first torpor using GENESIS[70]. We first fit a linear mixed model using the fixed effects and the genetic relatedness matrix as the random effect. We then performed SNP genotype association tests with 46,996 SNPS (MAF ≥ 0.05) and the fit from the linear mixed model. To correct for multiple hypothesis testing, we first estimated the number of independent SNP-association tests using the simpleM procedure[26]. We then applied a Bonferroni correction based upon the number of independent tests and a significance threshold of $\alpha = 0.1$. We considered any variant with a $p$-value $\leq 7.14 \times 10^{-6}$ to be significantly associated with the phenotype. As this was

an exploratory analysis, we identified additional variants suggestively associated with the phenotype by defining a lower significance threshold of $p = 2.13 \times 10^{-4}$, where we expected 10 variants to be significantly associated with the phenotype by random chance alone. To account for LD, we calculated the $r^2$ values for significant SNPs within the same scaffold using PLINK[71]. We removed those in moderate to high LD ($r^2 \geq 0.5$), reporting only the most significant variant.

We estimated heritability of the first day of torpor from the variance components of the linear mixed model. In addition, we also estimated heritability of this phenotype using a separate Bayesian mixed model with the MCMCglmm package in R[72]. Here we input the same fixed and random effects (i.e., genetic relatedness matrix) as in the linear mixed model. For the prior, we used an uninformative inverse-gamma distribution (with variance, V, set to 1 and belief parameter, nu, set to 0.002) on the variance components. We ran three chains, each with a total of 1,000,000 iterations, a burnin of 100,000 rounds and a thinning interval of 200 rounds. Here, all variables had Gelman-Rubin statistics of 1.00–1.01, with the absolute value of all autocorrelations <0.1 and effective sample sizes between 3682.8 and 5294.5. We combined the 3 chains in order to estimate the posterior mode and confidence intervals of the variance components.

Finally, we estimated the effects of the significant GWAS variants on the onset of torpor. Specifically, we used linear regression with the phenotype as the dependent variable and a matrix of significant variant genotypes, either with or without the fixed effects, as the explanatory variables. We also performed forward stepwise regression using genotype combinations from the top 10 significant variants.

**GWAS permutation testing and variant corroboration.** We used permutation testing to examine the empirical distribution of test statistics under the null model and to estimate the probability of detecting the observed number of variants at the suggestive threshold. We disrupted true genotype–phenotype associations by randomly reassigning the phenotype of each individual to another individual, while maintaining all other aspects of the dataset. We then performed the GWAS with permuted phenotypes using the same methods as described above. The GWAS permutation tests were repeated for a total of 5,000 rounds. We generated an empirical distribution of null $p$-values by rank ordering the values after each GWAS permutation. For each rank, we calculated the median $p$-value ± 97.5% confidence intervals. Additionally, after each round of GWAS permutation test, we counted the number of variants meeting the suggestive significance threshold (described above), and we generated an empirical distribution using the variant counts from all 5,000 rounds. The probability for observing the actual number of variants at the suggestive $p$-value threshold was estimated by determining the percentile at which 42 variants were detected using the Harrell–Davis distribution-free quantile estimator in the R package WRShd[73].

To corroborate variants identified as significantly and suggestively associated with hibernation onset, we used an alternative published method based on complementary pairs stability selection for genome-wide association, denoted ComPaSS-GWAS, where samples are split into complementary halves multiple times and GWAS are performed on the halves in each resample[36]. To compensate for the small sample size in our set, we randomly separated samples into two subsets, each two-thirds in size of the original sample pool. We allowed one-third of the samples to overlap between subsets and distributed the remaining two-thirds of samples into two complementary halves. Following the recommendations of ComPaSS-GWAS, we performed GWAS on the subsets for a total of 100 resamples, using a critical value of $\alpha = 1 \times 10^{-3}$ and a corroboration parameter of $\eta = 0.6$. We marked the variants in each resample as validated if the maximum $p$-value among both subsets was $p \leq 1 \times 10^{-3}$, and we considered variants corroborated for significance if they were validated in at least 60% of the resamples. Because the critical value recommended by ComPaSS-GWAS was identified empirically under assumptions of one million independent tests (i.e., a traditional GWAS critical value of $5 \times 10^{-8}$) and where samples were split into halves, we checked its appropriateness for use in our case by including 1,000 additional variants sampled uniformly across the distribution of effect sizes. At a type I error rate of 0.05, we expected no more than 50 of these additional variants to be corroborated for associated with the phenotype.

**Whole-genome resequencing and causal variant prioritization.** The whole genomes of $n = 6$ squirrels carrying both major alleles and $n = 6$ squirrels carrying at least one minor allele for the most significant variant, SNP 1, were sequenced with an average coverage of approximately 9× on an Illumina HiSeq 2500. Library preparation and sequencing were performed by Genewiz (South Plainfield, NJ). Read mapping and variant calling were performed as described above. We retained variants in moderate to high LD ($r^2 \geq 0.5$) with SNP 1; these were annotated and their effects predicted using SnpEff[74] with a custom database consisting of the Ensembl and NCBI gene annotations, as well as regions conserved across vertebrates, which were obtained using the UCSC Table Browser and lifted over from the hg38 assembly. Variants located within conserved elements were prioritized by their individual GERP++ scores[37], while those within or <1 kb from a gene annotation were prioritized by their predicted effects, with greatest priority being given to variants falling within both categories. We next analyzed for transcription factor binding site motifs that could be disrupted by the conserved (GERP++ >2) noncoding variants using Tomtom motif comparison software[38]. Here, the 15-bp

sequences surrounding each variant were input as motifs and searched against the JASPAR 2018 database. To further identify overlap within known regulatory elements, these variants were lifted over to hg37 and visualized using the ENCODE tracks in the UCSC browser. Finally, conserved noncoding variants that overlapped functional elements in hg37 were assessed for pathogenicity using FATHMM-MKL[39].

**eQTL analysis.** We applied an eQTL analysis to identify transcripts whose expression levels were significantly affected by the GWAS variants. Here, we used the EDGE-tag datasets from heart, liver, skeletal muscle (SkM) and brown adipose tissue (BAT) as previously published[16,40]. The squirrels assayed in these prior transcriptome studies were also genotyped in this study: heart ($n = 22$), liver ($n = 23$), SkM ($n = 22$) and BAT ($n = 43$). Due to small sample sizes, we limited our eQTL association tests to significant and suggestive GWAS SNPS with MAF ≥ 0.2 in heart, liver and SkM and ≥0.1 in BAT, which ensured that a minimum of 9 samples contained at least one minor allele.

Tests for both *cis*- (±500 kb) and *trans*-eQTLs were performed with Matrix eQTL[75] under an additive linear model. As the purpose of the original EDGE-tag studies was to identify differentially expressed transcripts among distinct physiological states within the hibernator's year, we included physiological state as a covariate (five states in heart, liver and SkM: spring cold, SpC; summer active, SA; interbout-aroused in hibernation, IBA; entering torpor in hibernation, Ent; and late torpor in hibernation, LT; nine states in BAT, in addition to those previously mentioned: spring warm, SW; fall transiton, FT; early torpor in hibernation, ET; early in arousal from torpor in hibernation, EAr). In BAT, sequencing platform was also included due to the observed count bias[16]. Finally, to control for outliers and following recommendations by Matrix eQTL, the counts for each Edge-tag were ranked and quantile-normalized before testing.

**Statistics and reproducibility.** Details about the different statistical analyses can be found in each of the subsections of the methods above. Unless otherwise stated, statistical analyses were performed in R[59]. Sample size was defined as number of ground squirrels genotyped by ddRAD sequencing in this study ($n = 153$); each squirrel served as an independent biological replicate. For the genome-wide association scan and heritability estimates, sample size was limited to the number of genotyped squirrels that also had phenotypic records of hibernation onset ($n = 72$). For the eQTL analyses, sample size was limited to the number of genotyped squirrels with published transcriptome sequence data, which varied by tissue ($n = 22$ for heart and skeletal muscle, $n = 23$ for liver and $n = 43$ for brown adipose tissue). Technical replicates were not used, although a subset of the genotyped squirrels ($n = 12$) were resequenced using whole-genome sequencing. Here, genotypes were assessed using an independent analysis.

**Reporting summary.** Further information on research design is available in the Nature Research Reporting Summary linked to this article.

## Data availability

Sequencing data from the ddRAD-seq and whole-genome resequencing experiments were deposited at the NCBI Sequence Read Archive (SRA) under project accession PRJNA420609. Sequencing data from the HiRise Genome assembly experiment was deposited under the NCBI BioProject accession PRJNA420392. The datasets detailing 13-lined ground squirrel genetic variation and the hibernation onset GWAS summary statistics are available at the following Open Science Framework repository: https://osf.io/a6v7w/?view_only=8d2841006be74207990f275de62d5436.

## Code availability

Custom software used to lift over gene annotations as described above is available at the following github repository: https://github.com/krgrabek/liftannot.

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

## Acknowledgements

This work was supported by the National Science Foundation grant 1642184 to C.D. Bustamante and the National Institutes of Health grant R01HL089049 to S.L. Martin. G.F.C was supported by the Brazilian Scientific Mobility Program - Ciência Sem Fronteiras Fellowship (CAPES Foundation, Brazil). The funders had no role in study design, data collection and analysis, decision to publish, or preparation of the manuscript. We thank members of the Bustamante Lab for their helpful discussion while this research was being conducted. We also thank R. Russell, A. Hindle and members of the Martin lab who assisted with the 13-lined ground squirrel care, surgical implantation of dataloggers and collection of tissue. C.D. Bustamante is a Chan Zuckerberg Biohub investigator.

## Author contributions

K.R.G. and C.D.B. conceived the project. L.E.E., D.K.M and S.L.M. collected samples and curated phenotypic data essential to the project. T.F.C. designed and provided technical support of the ddRAD-seq experiments and bioinformatic analysis. K.R.G., K.K.S., G.F.C., and S.C.S. generated and sequenced the sample libraries. K.R.G. analyzed and interpreted the data. C.D.B. supervised the project. K.R.G. drafted the manuscript. C.D.B. and S.L.M. revised the manuscript. All authors reviewed the manuscript.

## Competing interests

The authors declare no competing non-financial interests but the following competing financial interests: K.G. is co-founder of, equity owner in and chief scientific officer to Fauna Bio Incorporated. S.M. is an equity owner in and advisor to Fauna Bio Incorporated. C.B. is an equity owner in and advisor to both Fauna Bio Incorporated and Dovetail Genomics.
