## [Peer Review File · Communications Biology]

Reviewers' comments:

Reviewer #1 (Remarks to the Author):

In this manuscript, Grabek et al. report the first genetic analysis of a mammalian hibernation phenotype using 13-lined ground squirrels, which exhibit a strong circannual cycle that drives the timing of hibernation. They used an improved ground squirrel genome assembly and an extensive collection of data and samples to identify genes associated with the timing of hibernation onset via a genome wide association scan (GWAS). Their top GWAS hits account for a large amount of the variability in hibernation timing, and those candidate genes with known functions can be rationalized to have tissue-specific roles in the hibernation phenotype. Additionally, the authors find that the timing of hibernation onset exhibits high heritability. They identify a putative causal SNP in the promotor of their top GWAS hit (the poorly-characterized FAM204a) and use an eQTL approach to discover an association between their top SNPs and gene expression levels in the corresponding candidate genes.

I feel very positively about this manuscript. The entire field of hibernation biology has suffered due to a complete dearth of genetics; this manuscript serves to change that and it represents a significant advance in the field. To my knowledge, this is the first application of GWAS and eQTL approaches to any hibernation phenotype in any hibernator, and the authors' work has yielded the first list of genes that have a putative causative role in the timing of hibernation onset. This manuscript provides a starting point for experimental work in hibernation and a roadmap for other genetic studies in non-standard model organisms. This paper should be of broad interest to the hibernation community and beyond. The claims made by the authors are supported by their data and analysis, and I find no problems with the statistics. I have no major concerns but I do have some points for the authors to address:

1. The presentation of Fig.6 needs improvement:

- i. It is not reasonable to force the reader to look in the Methods to find the abbreviations for the panels in the middle-right and right-hand columns. This information should be included at the end of the legend. A full description of the sampling states should be provided in the Methods or (preferably) in the legend of a supplemental figure depicting said sampling times relative to season & body temp.
- ii. What are genotypes 0, 1, and 2 in the panels in left-hand columns, and how do these relate to the variants in Table 3? This is not explained anywhere.
- iii. I assume the colored dots in the two right-hand columns correspond to the genotypes in the two left columns? This was not readily apparent -- there is a lot going on here and it all needs explanation. Are the dots showing all individual observations that underlie the box and whisker plots?
- iv. What "sampling state" was used for the expression data in the middle-left column? If this is just a pooled version of the middle-right column data, I suggest adding a similar "pooled by genotype" panel for the trans-eQTL data so the reader can more easily make out the expression differences by genotype for the trans-eQTL genes. Perhaps put those panels into a far right column.
- v. A description of the summary statistics is lacking: what or how many samples are depicted in each panel?
- vi. Expression differences by sampling state (two right-hand columns) are never really addressed in the main text. What is the point of showing these panels? Explain the point to the reader or simplify the figure.

2. Line 337: "estimates for immersion into hibernation" -- should this be "immergence"?

Reviewer #2 (Remarks to the Author):

Grabek et al. is a forward-thinking study aimed to improve our understanding of how the phenotype of hibernation is controlled as a function of genomic architecture. To do this, the authors used a naturally hibernating mammal that has one of the better sequenced genomes – the thirteen lined ground squirrel. To accomplish this lofty aim the authors had to perform extensive ground work in terms of improving the contiguity of the genome and combining this with seasonal transcriptome data.

Genomes from a divergent group of mammalian hibernators have been sequenced with varying levels of coverage. The thirteen-lined ground squirrel genome has approximately 495X coverage. Combining this high coverage genome, extensive transcriptome data, and in vivo measurements of animal physiology, provided Grabek and colleagues with sufficient data to examine genomic architecture as a driver of the hibernation phenotype. After first increasing the contiguity of the sequenced genome, the authors compared their new and improved assembly with individual genomic sequences of 153 ground squirrels collected at precisely defined time points and body temperatures - many of which were measured by surgically implanted dataloggers. Using genome-wide association scans (GWAS) and eQTLs, they identified genetic variants significantly associated with the onset of autumn entrance into torpor.

This analysis revealed many loci that potentially regulate the onset of torpor. Of interest is the locus containing the gene for cholinergic receptor, muscarinic 2 (CHRM2). The increase in CHRM2 expression is logical because this receptor slows heart rate by slowing the speed of depolarization and reducing the atrial contractile force.

Overall, Grabek et al. is a landmark paper and the first step in showing that GWAS studies have the potential of uncovering genes that are important for the hibernation phenotype. As the genomes of other hibernators are better defined, this instrument will become a powerful approach to identify genes that control hibernation in mammals.

Reviewer #3 (Remarks to the Author):

I like this manuscript very much and believe that it should be published pretty much as is, but with a bit of "reframing." I hasten to add that I have not attempted to critique the analytical aspects in any detail given that the authors' expertise in functional genomics exceeds my own. Rather, I have read the paper more as a "next step" in our understanding of the genetic/genomic controls of hibernation, in this case in the 13-lined ground squirrel. The authors have provided an analysis that moves the field forward by combining comparative genomic data with detailed behavioral and physiological data collected over many years for a large (by non-model standards) colony of animals from a single species. They conclude that hibernation in these animals is triggered by a surprisingly small suite of loci with high levels of heritability. This appears to contradict the conventional wisdom that complicated phenotypes must be controlled by many genes of small effect. Thus, the results are certainly novel and are likely to garner attention far beyond the community of hibernation biologists.

That said, I do believe that the authors need reframe their prose such that the paper reads more as a presentation of hypotheses versus its current form which largely reads as a declaration of certainty. For example, the abstract begins with the assertion that "mammalian hibernation ... [is] timed by a circannual clock." This seems to imply that such a clock is known to be genetically mediated, is of known distribution among hibernators, and is generally acknowledged to be identifiable across different organismal groups. In fact, none of these things is true. Instead ... and this is where the hypotheses are most interesting and relevant to the biology of hibernation ... there is an emerging consensus among hibernation biologists that the phenotype appears to be largely controlled by endogenous features that are robust to environmental manipulations. Indeed, data are accumulating from obligate hibernators across a broad phylogenetic spectrum that show that these organisms will demonstrate at least some features of hibernation no matter how their environments are altered, and that these features will manifest on an annual clocklike schedule. The data supporting this emerging consensus are thus far relatively limited, and largely anecdotal, so to present the idea as a certainty (e.g., there IS a circannual clock) really gets the manuscript off on the wrong foot and seems to invite criticism.

My suggestions for improvement would thus include a more nuanced presentation of the circannual clock hypothesis with greater attention paid to the data (anecdotal and otherwise) that are emerging from other organismal groups. Thus, the overall effect would be to present the results of this study as more of an exploration of the possible genetic features of a clock, warranting further exploration in additional organismal groups. This could be accomplished with minimal revision but with the significant consequence of presenting the study appropriately as a very strong and positive step

towards understanding the underlying genetic mechanisms that drive what is a fascinating and extreme phenotype across a broad phylogenetic spectrum.

Anne Yoder

Reviewer #4 (Remarks to the Author):

Dr Grabek and co-authors report a genome-wide association study on 153 ground squirrels (*Ictidomys tridecemlineatus*) to study the contribution of genetics and onset of hibernation. The authors built upon and improved on the current *Ictidomys tridecemlineatus* reference genome using deep whole genome sequencing. From this experiment, the authors were able to detect and genotype 786,453 bi-allelic genetic variants. After stringent QC, and fully justifying excluding genetic variants with minor allele frequency <5%, the authors analyzed 46,996 genetic variants for onset of hibernation. They preset the threshold for experiment-wide significance at $P < 7.14 \times 10^{-6}$.

The study is beautifully designed and flawlessly executed in terms of technical skill. In particular, the sequencing, genotype calling, detection of familial relationships (this could confound GWAS), and using a linear mixed model specifying genetic relatedness as the random effect factor is conservative. This would be an excellent model study on GWAS in non-human organisms where the genome has yet to be fully catalogued. Of note, the authors observed that onset of hibernation is close to 100% heritable, an observation they support with data.

My concern is on the level of statistical significance for FAM204A and EXOC4/CHCHD3. Both were borderline significant at $\alpha = 0.1$. Using a more strict $\alpha = 0.05$ threshold will lead to $P = 3.57 \times 10^{-6}$ as the new threshold, thus rendering the 2 loci suggestively associated (but no longer experiment-wide significance). Ordinarily, these results would require replication before publication for robustness, especially on a background of near 100% heritability on the phenotype.

Our specific point-by-point response to each comment follows, with referee comments indicated in italics followed by our response in normal font.

Reviewer #1

1. *The presentation of Fig.6 needs improvement:*

i. It is not reasonable to force the reader to look in the Methods to find the abbreviations for the panels in the middle-right and right-hand columns. This information should be included at the end of the legend. A full description of the sampling states should be provided in the Methods or (preferably) in the legend of a supplemental figure depicting said sampling times relative to season & body temp.

ii. What are genotypes 0, 1, and 2 in the panels in left-hand columns, and how do these relate to the variants in Table 3? This is not explained anywhere.

iii. I assume the colored dots in the two right-hand columns correspond to the genotypes in the two left columns? This was not readily apparent -- there is a lot going on here and it all needs explanation. Are the dots showing all individual observations that underlie the box and whisker plots?

iv. What “sampling state” was used for the expression data in the middle-left column? If this is just a pooled version of the middle-right column data, I suggest adding a similar “pooled by genotype” panel for the trans-eQTL data so the reader can more easily make out the expression differences by genotype for the trans-eQTL genes. Perhaps put those panels into a far right column.

v. A description of the summary statistics is lacking: what or how many samples are depicted in each panel?

vi. Expression differences by sampling state (two right-hand columns) are never really addressed in the main text. What is the point of showing these panels? Explain the point to the reader or simplify the figure.

We agree that the presentation of Fig. 6 could be improved. Highlighting point vi of the referee’s comment, “*Expression differences by sampling state... are never really adressed in the main text. What is the point of showing these panels?*”, we agree that the main point intended for these panels – that genotype, rather than sampling state, accounts for changes in mRNA expression, was not easily interpretable in the figure. Because we already accounted for sampling state effects by their inclusion as a covariate in our analysis (and stated as such in the methods), and because of the above comments, we decided to simplify Fig. 6. In the revised figure, we removed the two right columns showing expression differences by sampling state to Supplementary Figure 13, with the sampling states defined in the corresponding legend. We also added the “pooled by genotype” column for the trans-eQTL data to Fig. 6 as suggested. We now provide more detailed information about the genotypes and the number of samples depicted in each panel in the legend for Fig. 6 (referee points ii and v).

2. Line 337: “estimates for immersion into hibernation” -- should this be “immergence”?

Yes, thank you, we edited this sentence to “estimates for immergence into hibernation”.

Reviewer #3

I do believe that the authors need reframe their prose such that the paper reads more as a presentation of hypotheses versus its current form which largely reads as a declaration of certainty. For example, the abstract begins with the assertion that "mammalian hibernation ... [is] timed by a circannual clock." This seems to imply that such a clock is known to be genetically mediated, is of known distribution among hibernators, and is generally acknowledged to be identifiable across different organismal groups. In fact, none of these things is true. Instead ... and this is where the hypotheses are most interesting and relevant to the biology of hibernation ... there is an emerging consensus among hibernation biologists that the phenotype appears to be largely controlled by endogenous features that are robust to environmental manipulations. Indeed, data are accumulating from obligate hibernators across a broad phylogenetic spectrum that show that these organisms will demonstrate at least some features of hibernation no matter how their environments are altered, and that these features will manifest on an annual clocklike schedule. The data supporting this emerging consensus are thus far relatively limited, and largely anecdotal, so to present the idea as a certainty (e.g., there IS a circannual clock) really gets the manuscript off on the wrong foot and seems to invite criticism.

My suggestions for improvement would thus include a more nuanced presentation of the circannual clock hypothesis with greater attention paid to the data (anecdotal and otherwise) that are emerging from other organismal groups. Thus, the overall effect would be to present the results of this study as more of an exploration of the possible genetic features of a clock, warranting further exploration in additional organismal groups. This could be accomplished with minimal revision but with the significant consequence of presenting the study appropriately as a very strong and positive step towards understanding the underlying genetic mechanisms that drive what is a fascinating and extreme phenotype across a broad phylogenetic spectrum.

We thank the referee for pointing out that across the broad phylogenetic distribution of hibernators, there is not yet a unified consensus on whether a genetically mediated circannual clock controls the timing of hibernation. However, in the context of sciurid rodents, we think there is ample evidence for a circannual clock timing hibernation and disagree that the data are simply anecdotal. We therefore have narrowed the scope of our statement to be specific to sciurid hibernators, and we have provided several appropriate references that support this statement.

Reviewer #4

My concern is on the level of statistical significance for FAM204A and EXOC4/CHCHD3. Both were borderline significant at $\alpha = 0.1$. Using a more strict $\alpha = 0.05$ threshold will lead to $P = 3.57 \times 10^{-6}$ as the new threshold, thus rendering the 2 loci suggestively associated (but no longer experiment-wide significance). Ordinarily, these results would require replication before publication for robustness, especially on a background of near 100% heritability on the phenotype.

We agree with the referee that ordinarily these results would require replication before publication. Nonetheless, the GWAS detailed in our manuscript was performed retrospectively on samples previously collected for use in other non-genetic studies. We chose to relax the significance threshold to $\alpha=0.1$ because of the exploratory nature of the analysis and because of our limited sample size. Since additional samples were not available to independently replicate our results, we instead conducted two alternative analyses; we believe the results of these further substantiate our findings of 2 loci significantly, and 12 loci suggestively, associated with the onset of autumn torpor.

In the first analysis, we increased the number of GWAS permutation tests from 100 to 5,000. We then used the results from these tests to generate an empirical distribution of variants meeting our suggestive significance threshold ($p=2.13 \times 10^{-4}$) under the null model. Significantly, from this distribution, we estimated a probability of just 2% for observing 42 variants (as detected in our GWAS) being randomly associated with the onset of autumn torpor. We document our findings in a new supplementary figure (Supplementary Figure 8).

More specifically, to corroborate our variant associations, we used an alternative published method for independent replication, denoted ComPaSS-GWAS¹. In this approach, samples are split into complementary halves multiple times and GWAS are performed on the halves in each resample. Because it was not feasible to perform GWAS on two halves of an already small ($n=72$) sample set, we compensated by randomly separating samples into two subsets, each two-thirds in size ($n=48$) of the original sample pool. We allowed one-third of the samples ($n=24$) to overlap between subsets and distributed the remaining two-thirds ($n=48$) of samples into two complementary ($n=24$) halves. We then used the stringent critical value of $\alpha=1 \times 10^{-3}$, as recommended by the ComPaSS-GWAS publication; this value was identified empirically under assumptions of one million independent tests (i.e. from a traditional GWAS critical value of 5×10^{-8}). Following the ComPaSS-GWAS recommendations, we performed GWAS on the subsets for a total of 100 resamples and marked the variants in each resample as validated if the maximum p-value among both subsets was $p \leq 1 \times 10^{-3}$. Finally, we considered variants corroborated for significance if they were validated in at least 60% of the resamples. From this analysis, we corroborated both SNP 1 and SNP 2 as being significantly associated with the onset of autumn torpor, as well as four of the suggestively associated variants: SNP3, SNP 7, SNP 8 and SNP 10. We provide the corroboration values for all of the significant and suggestive loci in a new column in Table 3. In contrast, when we sampled 1,000

additional variants across the distribution of effect sizes, none were corroborated for association with the phenotype. This contrast is documented in a new supplementary figure (Supplementary Figure 11) in our manuscript.

In sum, despite not being able to formally replicate our GWAS findings because of a lack of sample availability, we believe that these two new analyses adequately compensate, serving as independent replication to bolster our findings of specific loci being associated with the onset of autumn torpor. Combined with our results from the whole-genome resequencing and eQTL analyses, we believe that the likelihood for replicating these associations is high. We hope the results from this initial exploratory analysis will encourage more formal genetic mapping studies (with larger sample sizes) in the field of hibernation. We emphasize that this is the first study in hibernators to use this type of genetic mapping approach to identify genes associated with hibernation physiology; we believe that it has, even with the relatively limited sample set for GWAS, demonstrated its promise.

Reference:

- 1 Sabourin, J. A. *et al.* ComPaSS-GWAS: A method to reduce type I error in genome-wide association studies when replication data are not available. *Genetic epidemiology* **43**, 102-111, doi:10.1002/gepi.22168 (2019).

REVIEWERS' COMMENTS:

Reviewer #4 (Remarks to the Author):

The authors present a thorough rebuttal to my query regarding the preset alpha threshold (reported as 0.1) for experiment-wide significance.

I fully agree with the authors that based on the new data presented in Supplementary Figure 8, both SNP1 and SNP2 are clearly associated and would withstand further statistical scrutiny (in light of the limitations acknowledged by the authors). The additional analysis using ComPaSS-GWAS (Sabourin et al., 2019) provides further very persuasive evidence that the associations described by SNP1 and SNP2 with autumn torpor are not driven by any sample subgroup within the study dataset. Both show very clear positive results when analyzed using this alternate method.

(i would be more cautious in drawing firmer conclusions regarding SNPs 3, 7, 8 and 10 however, as the signal in the primary analysis was not as strong).

C.C. Khor

Our specific point-by-point response to each comment follows, with referee comments indicated in italics followed by our response in normal font.

Referee Comment: *I would be more cautious in drawing firmer conclusions regarding SNPs 3, 7, 8 and 10 however, as the signal in the primary analysis was not as strong.*

We agree with the reviewer that the signal in the primary analysis for SNPs 3, 7, 8 and 10 was not as strong as for SNPs 1 and 2, and that these SNPs should not be interpreted as being significantly associated with the onset of hibernation without further replication. We have therefore revised the discussion to better reflect these conclusions, positing that only the significant loci (SNPs 1 and 2) are associated with the onset of hibernation, while the remaining loci need further replication.

Original text:

“Additionally, we found that both the significantly associated variants, as well as four of the suggestively associated variants, were corroborated for association using an ad-hoc alternative approach for independent replication. Combined with evidence that these variants affect gene expression levels, we therefore posit that despite small sample size, both the significant, as well as a number of the suggestive loci, are indeed associated with the onset of torpor. The results of our study now warrant independent replication with a larger sample size.”

Revised text:

“Additionally, we found that both the significantly associated variants were corroborated for association using an alternative approach for independent replication. We therefore posit that despite small sample size, both the significant loci are indeed associated with the onset of torpor. For the remaining suggestively associated loci, the results of our study now warrant independent replication with a larger sample size.”